# Differentiable Meta-Learning of Bandit Policies

**Craig Boutilier**
Google Research

**Chih-Wei Hsu**
Google Research

**Branislav Kveton**
Google Research

**Martin Mladenov**
Google Research

**Csaba Szepesvári**
DeepMind / University of Alberta

**Manzil Zaheer**
Google Research

## Abstract

Exploration policies in Bayesian bandits maximize the average reward over problem instances drawn from some distribution $\mathcal{P}$. In this work, we *learn* such policies for an unknown distribution $\mathcal{P}$ using samples from $\mathcal{P}$. Our approach is a form of meta-learning and exploits properties of $\mathcal{P}$ without making strong assumptions about its form. To do this, we parameterize our policies in a differentiable way and optimize them by policy gradients, an approach that is pleasantly general and easy to implement. We derive effective gradient estimators and propose novel variance reduction techniques. We also analyze and experiment with various bandit policy classes, including neural networks and a novel softmax policy. The latter has regret guarantees and is a natural starting point for our optimization. Our experiments show the versatility of our approach. We also observe that neural network policies can learn implicit biases expressed only through the sampled instances.

## 1 Introduction

A *stochastic bandit* [36, 9, 38] is an online learning problem where a *learning agent* sequentially pulls arms with stochastic rewards. The agent aims to maximize its expected cumulative reward over some horizon. It does not know the mean rewards of the arms a priori and learns them by pulling the arms. This induces the well-known *exploration-exploitation trade-off*: *explore*, and learn more about an arm; or *exploit*, and pull the arm with the highest estimated reward. In a clinical trial, the *arm* might be a treatment and its *reward* is the outcome of that treatment for a patient.

Bandit algorithms are typically designed to have *low regret*, worst-case or instance-dependent, for some problem class of interest to the algorithm designer [38]. While regret guarantees are reassuring, this approach often results in algorithms that are overly conservative, since they do not exploit the full properties of the problem class or objective. We explore an alternative view, which is to *learn* a bandit algorithm. Specifically, we assume that the agent has access to bandit instances sampled from an unknown distribution $\mathcal{P}$ and attempts to learn a bandit algorithm with a high *Bayes reward*, the average reward over the instances drawn from $\mathcal{P}$. In essence, we automate the learning of policies for *Bayesian bandits* [16]. Our approach can be viewed as a form of *meta-learning* [55, 56, 11, 12] with gradient ascent [25].

A classic approach to Bayesian bandits is to design Bayes optimal policies [27, 28], which take a simple form for specific priors $\mathcal{P}$. Our approach is more general, since it makes minimal assumptions about $\mathcal{P}$ and optimized policies. It is also more computationally efficient and easier to parallelize. However, we lose guarantees on Bayes optimality. Another line of work [48, 59, 49] bounds the Bayes regret of classic bandit policies. These policies also have instance-dependent regret bounds and thus are more conservative than our work, where we directly optimize the Bayes reward.

Overall, our aim is to make learning of bandit policies as straightforward as applying gradient descent to supervised learning problems. We take the following steps toward this goal. First, we carefully

formulate the problem of policy-gradient optimization of the Bayes reward of bandit policies. Second, we derive the reward gradient and propose novel baseline subtraction methods that reduce the variance of its empirical estimate. These methods are tailored to the bandit structure of our problem and are critical to making our approach practical. Third, we show how to differentiate several softmax bandit policies: Exp3, SoftElim, and neural networks with a softmax output layer. SoftElim is a new algorithm where the probability of pulling an arm is directly parameterized. We prove that its $n$-round regret is sublinear in $n$ for any $K$-armed bandit, as in UCB1 [9] and Thompson sampling (TS) [54, 3]. However, unlike UCB1 and TS, SoftElim is easy to optimize. Finally, we evaluate our methodology empirically on a range of bandit problems, highlighting the versatility of our approach. We also show that neural network policies can learn interesting biases encoded in the prior distribution $\mathcal{P}$.

## 2 Setting

This section introduces Bayesian bandits [27, 16]. To ease exposition, we first introduce multi-armed bandits [36, 9, 38]. To simplify notation, we define $[n] = \{1, \dots, n\}$.

### 2.1 Stochastic Multi-Armed Bandit

A *stochastic multi-armed bandit* [36, 9, 38] is an online learning problem where a learning agent interacts with the environment by pulling $K$ arms in $n$ rounds. In round $t \in [n]$, the agent *pulls* arm $I_t \in [K]$ and observes its reward. We denote the *reward* of arm $i$ in round $t$ by $Y_{i,t}$ and the rewards of all arms in round $t$ by $Y_t = (Y_{1,t}, \dots, Y_{K,t})$. We assume that the rewards are drawn i.i.d. over rounds as $Y_t \sim P$, where $P$ is a joint probability distribution over rewards of all arms, with support $[0, 1]^K$. Since $P$ fully characterizes the bandit problem, we call it a *problem instance*. We denote the realized rewards in all $n$ rounds by $Y = (Y_t)_{t=1}^n \in [0, 1]^{K \times n}$.

At a high level, all agents for solving bandit problems can be viewed as policies that map the history of the agent to the pulled arm. The *history* of the agent at the beginning of round $t$ is

$$H_t = (I_1, \dots, I_{t-1}, Y_{I_1,1}, \dots, Y_{I_{t-1},t-1}),$$

which is a vector of pulled arm indices and corresponding observations up to round $t$. The agent is a *randomized policy* $\pi_w(\cdot \mid H_t) \in \Delta_K$, where $\Delta_K$ is a $K$-dimensional probability simplex, $w \in \mathcal{W}$ are *policy parameters*, and $\mathcal{W}$ is the set of *feasible policy parameters*. In this notation, $\pi_w(i \mid H_t)$ is the probability that arm $i$ is pulled in round $t$ given history $H_t$ and $I_t \sim \pi_w(\cdot \mid H_t)$. To simplify notation, we write $\pi$ instead of $\pi_w$ when the dependence on $w$ does not have to be stressed. We also define $I_{i:j} = (I_\ell)_{\ell=i}^j$ and $I = I_{1:n}$.

The *expected $n$-round reward* of policy $\pi$ in problem instance $P$ is $r(n, P; \pi) = \mathbb{E}\left[\sum_{t=1}^n Y_{I_t,t} \mid P\right]$. Note that the expectation is over both realized rewards $Y$ and pulled arms $I$. A typical goal is to maximize $r(n, P; \pi)$, which is equivalent to regret minimization. The *expected $n$-round regret* of policy $\pi$ is defined as $R(n, P; \pi) = \mathbb{E}\left[\sum_{t=1}^n Y_{i_*(P),t} - Y_{I_t,t} \mid P\right]$, where $i_*(P) = \arg\max_{i \in [K]} \mu_i$ is the *optimal arm* in problem instance $P$ with mean arm rewards $\mu = \mathbb{E}[Y_t \mid P]$ for any $t \in [n]$.

### 2.2 Bayesian Bandit

A *Bayesian bandit* [27, 16] is a bandit where a learning agent interacts with problem instances $P$ drawn i.i.d. from a known prior distribution $\mathcal{P}$. In this work, the problem instance $P$ is the $K$-armed bandit in Section 2.1. The learning agent interacts as follows. First, the problem instance is drawn as $P \sim \mathcal{P}$ and all rewards in it are sampled as $Y \sim P$. Then the agent starts interacting with instance $P$ for $n$ rounds, starting from round 1. The agent does not know $P$ or $Y$.

The quality of Bayesian bandit policies is measured by their $n$-round Bayes reward and regret. The *$n$-round Bayes reward* of policy $\pi$ is

$$r(n; \pi) = \mathbb{E}[r(n, P; \pi)] = \mathbb{E}\left[\sum_{t=1}^n Y_{I_t,t}\right]. \tag{1}$$

When compared to Section 2.1, the expectation is also over problem instances $P$. Our goal is to maximize $r(n; \pi_w)$ with respect to $w$. This is equivalent to minimizing the *$n$-round Bayes regret*

$$R(n; \pi) = \mathbb{E}[R(n, P; \pi)] = \mathbb{E}\left[\sum_{t=1}^n Y_{i_*(P),t} - Y_{I_t,t}\right], \tag{2}$$

---

**Algorithm 1** Gradient-based optimization of bandit policies.

---

1: **Inputs:** Policy parameters $w_0 \in \mathcal{W}$, number of iterations $L$, learning rate $\alpha$, and batch size $m$

2: $w \leftarrow w_0$
3: **for** $\ell = 1, \ldots, L$ **do**
4:     **for** $j = 1, \ldots, m$ **do**
5:         Sample $P^j \sim \mathcal{P}$; sample $Y^j \sim P^j$; and apply policy $\pi_w$ to $Y^j$ to get $I^j$
6:     Let $\hat{g}(n; \pi_w)$ be an estimate of $\nabla_w r(n; \pi_w)$ from $(Y^j)_{j=1}^m$ and $(I^j)_{j=1}^m$
7:     $w \leftarrow w + \alpha \, \hat{g}(n; \pi_w)$

8: **Output:** Learned policy parameters $w$

---

where $i_*(P)$ is defined as in Section 2.1.

To clarify our setting, we give examples of a Bayesian bandit and a policy for solving it below.

**Example 1** (Bernoulli bandit with a uniform beta prior). *Let $\mu \in [0, 1]^K$ be a vector of arm means and $y \in [0, 1]^K$ be a vector of realized rewards. Then the reward distribution is $P(y) = \prod_{i=1}^K \mathrm{Ber}(y_i; \mu_i)$ and the prior distribution over arm means is $\mathcal{P}(\mu) = \prod_{i=1}^K \mathrm{Beta}(\mu_i; 1, 1)$.*

**Example 2** ($\varepsilon$-greedy policy). *The algorithm is parameterized by $w \in [0, 1]$ and works as follows. It pulls the best empirical arm with probability $1 - w$ and a random arm with probability $w$. That is,*

$$\pi_w(i \mid H_t) = (1 - w)\mathbb{1}\left\{i = \arg\max_{j \in [K]} \hat{\mu}_{j,t-1}\right\} + w/K \,,$$

*where $\hat{\mu}_{i,t}$ is the empirical mean of arm $i$ after $t$ rounds.*

The bandit policy $\pi_w$ in Example 2 and our objective in (1) showcase the two-level character of our optimization problem. At the lower level, $\pi_w$ adapts to an unknown problem instance $P$, because it is a function of the history $H_t$ that depends on $P$. At the higher level, we optimize (1) through $w$, so that $\pi_w$ adapts to $P$ as efficiently as possible, on average over $P \sim \mathcal{P}$.

## 3 Bandit Policy Optimization

We propose a general and data-dependent algorithm for learning bandit policies. The key idea is to maximize the Bayes reward $r(n; \pi)$ by gradient ascent on sampled problem instances from $\mathcal{P}$. Our proposed algorithm, which we call `GradBand`, is presented in Algorithm 1. `GradBand` is initialized with policy parameters $w_0$. In each iteration, parameters of the last policy $w$ are updated by gradient ascent using $\hat{g}(n; \pi_w)$, an empirical estimate of the *reward gradient* at the last policy, $\nabla_w r(n; \pi_w)$. To compute $\hat{g}(n; \pi_w)$, we run the policy $\pi_w$ on $m$ sampled problem instances from $\mathcal{P}$. We denote the $j$-th sampled instance by $P^j$, its realized rewards by $Y^j \in [0, 1]^{K \times n}$, and the corresponding pulled arms by $I^j \in [K]^n$.

The per-iteration time complexity of `GradBand` is $O(Kmn)$, because in each iteration we sample $m$ problem instances from $\mathcal{P}$ with horizon $n$ and $K$ arms, and run a policy in each of them. Technically speaking, $\hat{g}(n; \pi_w)$ depends only on $Y^j$ and $I^j$, and $I^j$ depends on $P_j$ only through $Y^j$. Therefore, `GradBand` does not need to know the instances $P_j$ or their distribution $\mathcal{P}$ to compute the gradient.

`GradBand` is general, data-dependent, and directly optimizes the Bayes reward $r(n; \pi)$. However, since $r(n; \pi)$ is a complex function of the adaptive bandit policy $\pi$ and environment, it is difficult to provide meaningful guarantees on optimizing it. This is one reason why existing bandit papers analyze theoretically manageable regret upper bounds and not the regret. We provide the first such guarantee below.

**Theorem 1.** *Consider a 2-armed Gaussian bandit where the reward of arm $i$ in round $t$ is $Y_{i,t} \sim \mathcal{N}(\mu_i, 1)$. Consider an* explore-then-commit *policy $\pi_h$ with parameter $h \in \mathcal{W} = [1, \lfloor n/2 \rfloor]$ that explores each arm $\bar{h} = \lfloor h \rfloor + Z$ times for $Z \sim \mathrm{Ber}(h - \lfloor h \rfloor)$. Then for any prior distribution $\mathcal{P}$ over arm means $\mu \in \mathbb{R}^2$, we have that $r(n; \pi_h)$ is concave in $h$.*

The claim is proved in Appendix A. The key insight is that $r(n; \pi_h)$ of the explore-then-commit policy in a 2-armed Gaussian bandit has a closed form, differentiable with respect to $h$. The randomization

in Theorem 1 is only needed to extend the policy to continuous exploration horizons $h$. In this case, `GradBand` enjoys the same convergence guarantees as gradient descent for convex functions.

`GradBand` is a meta-algorithm. To fully exploit its power, we must specify the policy $\pi$ and be able to compute its empirical gradient $\hat{g}(n; \pi)$. In Section 4, we derive the gradient and show how to reduce its variance. In Section 5, we study several differentiable bandit policies. Before we proceed, we relate our objective and algorithm design to prior work.

**Stochastic multi-armed bandits:** Our objective, the maximization of $\mathbb{E}\left[\sum_{t=1}^n Y_{I_t,t}\right]$, differs from maximizing $\mathbb{E}\left[\sum_{t=1}^n Y_{I_t,t} \mid P\right]$ in any problem instance $P$, which is standard in bandits [36, 9, 38]. The latter objective is more demanding, since it requires optimizing equally for likely and unlikely instances $P \sim \mathcal{P}$. Our objective is more appropriate when $\mathcal{P}$ can be estimated from data and the average reward is preferred to guarding against worst-case failures.

**Bayesian bandits:** Early works on Bayesian bandits [27, 16, 28] derived Bayes optimal policies that required specific conjugate priors $\mathcal{P}$. We do not make any such assumptions on $\mathcal{P}$. However, we do lose Bayes optimality guarantees, as the optimal policy may not have the same form as $\pi$. Since `GradBand` differentiates policies, it can be computationally costly. Nevertheless, it is less costly and easier to parallelize than the computation of typical Bayes optimal policies (Section 6.2).

**Reinforcement learning:** Learning of policy $\pi$ is also an instance of *reinforcement learning (RL)* [51], where the *state* in round $t$ is history $H_t$, the *action* is the pulled arm $I_t$, and the *reward* is the reward of the pulled arm $Y_{I_t,t}$. The main challenge is that the number of dimensions in $H_t$ increases linearly with round $t$. So any RL method that solves this problem must introduce some structure to deal with the *curse of dimensionality*. Since it is not clear what the shape of the value function might be, we opt for optimizing parametric bandit policies (Section 5) by policy gradients [60]. The main novelty in our application of policy gradients are baseline subtraction techniques that are tailored to the bandit structure of our problem.

## 4 Reward Gradient

Using relatively standard techniques, we show that the reward gradient has the following form.

**Theorem 2.** *Let* $b_t : [K]^{t-1} \times [0,1]^{K \times n} \to \mathbb{R}$ *be any function of previous* $t-1$ *pulled arms and all realized rewards, for any round* $t \in [n]$. *Then*

$$\nabla_w r(n; \pi_w) = \sum_{t=1}^n \mathbb{E}\left[\nabla_w \log \pi_w(I_t \mid H_t) \left(\sum_{s=t}^n Y_{I_s,s} - b_t(I_{1:t-1}, Y)\right)\right].$$

The claim is proved in Appendix B. The collection of functions $b = (b_t)_{t=1}^n$ in Theorem 2 is known as a *baseline* [60, 53] and its purpose is to reduce the variance of empirical gradient estimates. The baseline does not change the gradient, as long as it does not depend on future actions taken by policy $\pi$. In our case, this means that $b_t$ can be any function of all past pulls $I_{1:t-1}$, all realized rewards $Y$, problem instance $P$, and policy parameters $w$; although we do not make the last two explicit. The empirical gradient, for $m$ sampled instances in `GradBand`, is

$$\hat{g}(n; \pi_w) = \frac{1}{m} \sum_{j=1}^m \sum_{t=1}^n \nabla_w \log \pi_w(I_t^j \mid H_t^j) \left(\sum_{s=t}^n Y_{I_s^j,s}^j - b_t(I_{1:t-1}^j, Y^j)\right), \tag{3}$$

where $j$ indexes the $j$-th random experiment in `GradBand`.

Existing variance minimizing techniques (Section 7) are hard to apply to our problem because our state space $H_t$ is at least exponential in $n$; and the shape of the value function, the future regret as a function $H_t$, is unknown and likely hard to approximate. Therefore, we propose our own baselines.

Our first baseline is *no baseline* $b_t^{\text{NONE}}(I_{1:t-1}, Y) = 0$. Not surprisingly, $b^{\text{NONE}}$ performs poorly, even when learning bandit policies at short horizons (Section 6.2).

Our second baseline is $b_t^{\text{OPT}}(I_{1:t-1}, Y) = \sum_{s=t}^n Y_{i_*(P),s}$, where $i_*(P)$ is the best arm in instance $P$, as defined in (2). This baseline is suitable for analyzable bandit policies. Specifically, if the policy has a sublinear regret with a high probability for any $P$, we have $\sum_{t=1}^n Y_{i_*(P),t} - Y_{I_t,t} = o(n)$ and hence $b_t(I_{1:t-1}, Y) - \sum_{s=t}^n Y_{I_s,s} = o(n)$ (Theorem 2) for any $s \in [n]$, both with a high probability for any $P$.

One limitation of $b^{\text{OPT}}$ is that the best arm may be unknown, for instance if `GradBand` was only given sampled realized rewards $Y^j$ but not sampled instances $P^j$. This motivates our third baseline, which is the reward of an independent run of policy $\pi$. Let $(J_t)_{t=1}^n$ be the pulled arms in an independent run of $\pi$. Then $b_t^{\text{SELF}}(I_{1:t-1}, Y) = \sum_{s=t}^n Y_{J_s,s}$. Similarly to $b^{\text{OPT}}$, $b^{\text{SELF}}$ is suitable for any policy that concentrates on a single arm over time. Unlike $b^{\text{OPT}}$, it does not rely on knowing the best arm.

# 5  Differentiable Bandit Algorithms

We assume that $\nabla_w \log \pi_w(I_t \mid H_t)$ in Theorem 2 exists, so that the bandit policy is differentiable. Unfortunately, it seems that only a few bandit policies satisfy this assumption. For instance, UCB algorithms [9, 20, 1] are not differentiable because $\pi_w(i \mid H_t) \in \{0, 1\}$ is a step function. While Thompson sampling [54, 3, 4] is randomized, $\pi_w(i \mid H_t)$ is induced by a hard maximization over randomized posterior mean rewards. Therefore, a unique gradient may not exist. Even if it does, $\pi_w(i \mid H_t)$ does not have a closed form and thus is hard to differentiate computationally efficiently.

In the rest of this section, we introduce three softmax designs that can be differentiated analytically and derive a gradient for each of them. To simplify notation, we use $\pi_{i,t} = \pi_w(i \mid H_t)$. We note that the $\varepsilon$-greedy policy [52] and Boltzmann exploration [52, 18] are also differentiable, although we do not study them here.

## 5.1  Algorithm `Exp3`

`Exp3` [8] is a non-stochastic bandit algorithm that pulls arm $i$ in round $t$ with probability

$$\pi_{i,t} = (1-w)\exp[\eta S_{i,t}]\Big/ \sum_{j=1}^K \exp[\eta S_{j,t}] + w/K \,, \tag{4}$$

where $S_{i,t} = \sum_{\ell=1}^{t-1} \mathbb{1}\{I_\ell = i\} \pi_{i,\ell}^{-1} Y_{i,\ell}$ is the inverse propensity score estimate of the cumulative reward of arm $i$ in the first $t-1$ rounds, $\eta$ is a learning rate, and $w$ is a parameter that guarantees sufficient exploration. When rewards are $[0, 1]$, `Exp3` attains $O(\sqrt{Kn})$ regret when $\eta = w/K$ and $w = \min\left\{1, \sqrt{K \log K}/\sqrt{(e-1)n}\right\}$. In this work, we *optimize* $w$ by `GradBand`. When $\eta$ is set as above, we get the following gradient.

**Lemma 3.** *Define $\pi_{i,t}$ as in (4). Let $\eta = w/K$, $V_{i,t} = \exp[w S_{i,t}/K]$, and $V_t = \sum_{j=1}^K V_{j,t}$. Then*

$$\nabla_w \log \pi_{i,t} = \frac{1}{\pi_{i,t}} \left[ \frac{V_{i,t}}{V_t} \left[ (1-w) \left[ \frac{S_{i,t}}{K} - \sum_{j=1}^K \frac{V_{j,t}}{V_t} \frac{S_{j,t}}{K} \right] - 1 \right] + \frac{1}{K} \right] \,.$$

The lemma is proved in Appendix B. Although `Exp3` is differentiable, it is conservative in stochastic problems, even after we optimize $w$ (Section 6). Thus we propose a new algorithm `SoftElim`.

## 5.2  Algorithm `SoftElim`

Our bandit algorithm works as follows. Let $\hat{\mu}_{i,t}$ be the empirical mean of arm $i$ after $t$ rounds and $T_{i,t}$ be the number of pulls of arm $i$ after $t$ rounds. Each arm is initially pulled once. Then, in round $t > K$, arm $i$ is pulled with probability

$$\pi_{i,t} = \exp[-S_{i,t}/w^2]\Big/ \sum_{j=1}^K \exp[-S_{j,t}/w^2] \,, \tag{5}$$

where $S_{i,t} = 2\left(\max_{j\in[K]} \hat{\mu}_{j,t-1} - \hat{\mu}_{i,t-1}\right)^2 T_{i,t-1}$ is the *score* of arm $i$ in round $t$ and $w > 0$ is a tunable *exploration parameter*. Since $S_{i,t} \geq 0$ and $\pi_{i,t} \propto \exp[-S_{i,t}/w^2]$, higher values of $w$ lead to more exploration. Also note that $\exp[-S_{i,t}/w^2] \in [0, 1]$. Therefore, our algorithm can be viewed as "soft" elimination [7] of arms with high scores and we call it `SoftElim`.

The key idea in the design of `SoftElim` is that an arm is unlikely to be pulled if it has been pulled "often" and its empirical mean is low relatively to the highest empirical mean. This follows from the definition of score $S_{i,t}$. More interestingly, when a suboptimal arm has been pulled "often" and has the highest empirical mean, the optimal arm is pulled proportionally to how much its empirical mean deviates from the actual mean. This latter property implies sufficient optimism and leads to a regret analysis (Appendix C).

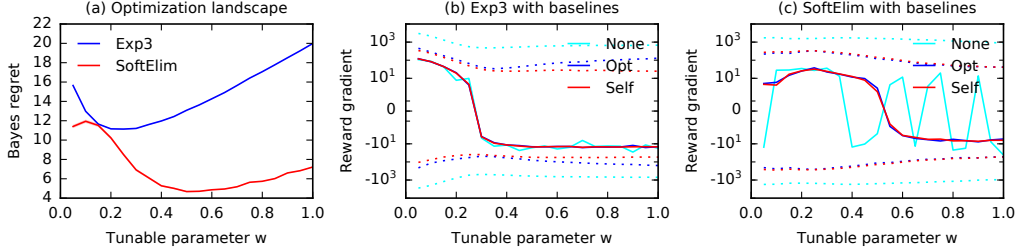

Figure 1: The Bayes regret of `Exp3` and `SoftElim`, and the corresponding reward gradients. In the last two plots, the solid lines are estimated reward gradients for $m = 10\,000$ in (3) and the dotted lines mark high-probability regions for $m = 1$.

Since $\log \pi_{i,t} = -w^{-2} S_{i,t} - \log \sum_{j=1}^{K} \exp[-S_{j,t}/w^2]$, we have

$$\nabla_w \log \pi_{i,t} = 2w^{-3} \left( S_{i,t} - \sum_{j=1}^{K} S_{j,t} \exp[-S_{j,t}/w^2] \Big/ \sum_{j=1}^{K} \exp[-S_{j,t}/w^2] \right).$$

Therefore, `SoftElim` can be easily differentiated and optimized by `GradBand`. `SoftElim` also has a sublinear regret in any problem instance, as we show below.

**Theorem 4.** *Let $P$ be any $K$-armed bandit problem where arm $1$ is optimal, that is $\mu_1 > \max_{i>1} \mu_i$. Let $\Delta_i = \mu_1 - \mu_i$ and $w = \sqrt{8}$. Then $R(n, P; \pi_w) \le \sum_{i=2}^{K} (2e + 1) \left( 16\Delta_i^{-1} \log n + \Delta_i \right) + 5\Delta_i$.*

The proof of Theorem 4 is sketched in Appendix C.1 and presented in Appendix C.2. The value of $w = \sqrt{8}$ is obtained by tuning. An analogous bound, with worse constants, can be derived for any $w \in (1, \sqrt{8}]$. This can be seen from the proof, which only requires that $\gamma = w^{-2} \in [1/8, 1)$. Finally, note that our bound depends on gaps $\Delta_i$ and $\log n$ as `UCB1` [9]. Thus it is near optimal.

### 5.3 Recurrent Neural Network

Now we take designs (4) and (5) a step further. In particular, as both are softmaxes on hand-crafted features, which allow theoretical analysis, we attempt to *learn the features* using a *recurrent neural network (RNN)*. The RNN works as follows. In round $t$, it takes arm $I_t$ and reward $Y_{I_t,t}$ as inputs, updates its state $s_t$, and outputs the probability $\pi_{i,t+1}$ of pulling each arm $i$ for round $t + 1$. That is,

$$s_t = \mathrm{RNN}_\Phi(s_{t-1}, (I_t, Y_{I_t,t})), \quad \pi_{i,t+1} = \exp[v_i^\top s_t] \Big/ \sum_{j=1}^{K} \exp[v_j^\top s_t].$$

The optimized parameters $w = (\Phi, \{v_i\}_{i=1}^{K})$ are the RNN parameters $\Phi$ and per-arm parameters $v_i$. The aim for the RNN is to learn to track suitable sufficient statistics through its internal state $s_t$. That state is initialized at $s_0 = \mathbf{0}$. Our RNN is an LSTM [30] with a $d$-dimensional latent state. We assume that the rewards are Bernoulli. The details of our implementation are in Appendix E.

## 6 Experiments

To demonstrate the generality and efficacy of our approach to learning bandit policies, we conduct four experiments. In Section 6.1, we study the reward gradient and its variance in a simple problem. In Section 6.2, we optimize `Exp3` and `SoftElim` policies. In Section 6.3, we study the sensitivity of our approach to model and algorithm parameters. In Section 6.4, we optimize RNN policies.

The performance of policies is measured by the Bayes regret instead of the Bayes reward, since it directly indicates how near-optimal a policy is. Note that optimizing either optimizes the other. The regret is estimated from $1\,000$ i.i.d. sampled problem instances from $\mathcal{P}$, which are independent of those used by `GradBand`. The shaded areas in plots represent standard errors of our estimates. Our experiments are implemented in TensorFlow and PyTorch, on 112 cores and with 392 GB RAM.

### 6.1 Reward Gradient

Our first experiment is on a Bayesian bandit with $K = 2$ arms, Bernoulli reward distributions, and a horizon of $n = 200$ rounds. The prior distribution $\mathcal{P}$ is over two problem instances, $\mu = (0.6, 0.4)$

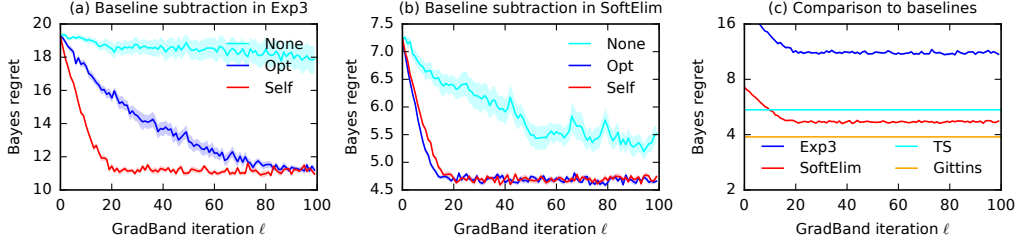

Figure 2: The Bayes regret of `Exp3` and `SoftElim` policies, as a function of `GradBand` iterations. All experiments are on a Bernoulli bandit with $K = 2$ arms (Section 6.2). We report the average regret over 20 runs.

and $\mu = (0.4, 0.6)$, both with probability 0.5. The bandit policy $\pi_w$ tries to identify the optimal arm, and thus to distinguish the instances, without knowing the instance that it interacts with. In this case, maximizing (1) with respect to $w$ is akin to achieving this as fast as statistically possible, without incurring high regret on average. The symmetry of the problem instances is unimportant.

The Bayes regret of `Exp3` and `SoftElim`, as a function of their parameter $w$, is shown in Figure 1a. Both are unimodal in $w$ and suitable for optimization by `GradBand`. `SoftElim` has a lower regret than `Exp3` for all $w$. In fact, the minimum regret of `Exp3` is higher than that of `SoftElim` without tuning ($w = 1$). The reward gradients of `Exp3` and `SoftElim` are reported in Figures 1b and 1c, respectively. We observe that baselines $b^{\text{OPT}}$ and $b^{\text{SELF}}$ lead to orders of magnitude lower variance than no baseline $b^{\text{NONE}}$. The variance of `SoftElim` gradients with $b^{\text{OPT}}$ and $b^{\text{SELF}}$ is comparable, while the variance of `Exp3` gradients with $b^{\text{SELF}}$ is two orders of magnitude lower for higher values of $w$.

## 6.2 Policy Optimization

In the second experiment, we apply `Exp3` and `SoftElim` to the problem in Section 6.1. The policies are optimized by `GradBand` with $w_0 = 1$, $L = 100$ iterations, learning rate $\alpha = c^{-1}L^{-\frac{1}{2}}$, and batch size $m = 1\,000$. The constant $c$ is chosen automatically so that $\|\hat{g}(n; \pi_{w_0})\| \le c$ holds with a high probability, to avoid manual tuning of the learning rate in our experiments.

In Figure 2a, we optimize `Exp3` with all baselines. With $b^{\text{SELF}}$, `GradBand` learns a near-optimal policy in fewer than 20 iterations. This is consistent with Figure 1b, where $b^{\text{SELF}}$ has the lowest variance. In Figure 2b, we optimize `SoftElim`; and the performance with $b^{\text{OPT}}$ and $b^{\text{SELF}}$ is comparable. This consistent with Figure 1c, where the variances of $b^{\text{OPT}}$ and $b^{\text{SELF}}$ are comparable. We conclude that $b^{\text{SELF}}$ is the best baseline overall and use it in all remaining experiments.

To evaluate the quality of our policies, we compare them to four well-known bandit policies: `UCB1` [9], Bernoulli `TS` [3] with $\text{Beta}(1, 1)$ prior, `UCB-V` [6], and the Gittins index [27]. These benchmarks are ideal points of comparison: (i) `UCB1` is arguably the most popular multi-armed bandit algorithm for $[0, 1]$ rewards. (ii) Bernoulli `TS` is near-optimal in Bernoulli bandits, which we use often in our experiments. We use randomized Bernoulli rounding [3] to apply `TS` to $[0, 1]$ rewards. (iii) `UCB-V` adapts the sub-Gaussian parameter of reward distributions based on observed rewards. This is similar to optimizing $w$ in `SoftElim`. (iv) The Gittins index is the optimal solution to our problem, under the assumption that the mean arm rewards are drawn i.i.d. from $\text{Beta}(1, 1)$. Finally, we also compare to the Dopamine [14] implementation of DQN [45] where the state is a concatenation of the following statistics for each arm: the number of observed ones, the number of observed zeros, the logarithm of both counts incremented by 1, the empirical mean, and a constant bias term.

The Bayes regret of our benchmarks is $9.95 \pm 0.03$ (`UCB1`), $5.47 \pm 0.05$ (`TS`), $15.79 \pm 0.03$ (`UCB-V`), $3.89 \pm 0.07$ (Gittins index), and $16.81 \pm 1.05$ (DQN). The regret of `SoftElim` is $4.74 \pm 0.03$, and falls between those of `TS` and the Gittins index. The regret of `Exp3` is $10.96 \pm 0.16$, which is not competitive. The regret of our policies and most competitive baselines is shown in Figure 2c. We conclude that tuned `SoftElim` outperforms a strong baseline, `TS`; and performs almost as well as the Gittins index. We note that the Gittins index provides the optimal solution in limited settings, like Bernoulli bandits, but even there it is computationally costly. For instance, our computation of the Gittins index for horizon $n = 200$ took almost two days. In comparison, tuning of `SoftElim` by `GradBand` takes about 20 seconds.

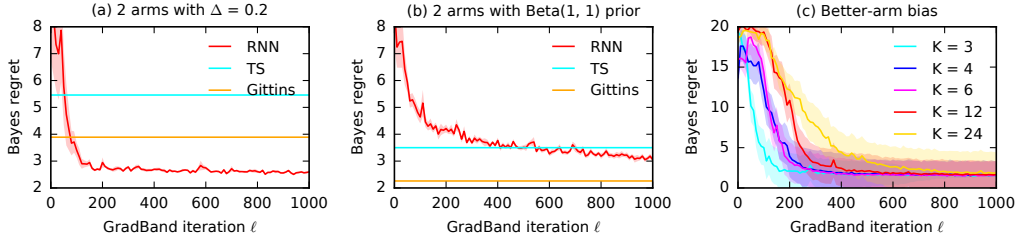

Figure 3: The Bayes regret of RNN policies, as a function of `GradBand` iterations. We report the average over 10 runs in the first two plots and the median in the last. The median excludes a few failed runs, which would skew the average.

Now we discuss failures of some benchmarks. `UCB-V` fails because its variance optimism induces too much initial exploration. This is harmful for somewhat short learning horizons in our experiments. DQN policies are unstable and require significant tuning to learn policies that outperform random actions; and still perform poorly. This is in a stark contrast with the simplicity of `GradBand`, which learns near-optimal policies using gradient ascent. In the remaining experiments, we only discuss the most competitive benchmarks, the Gittins index and `TS`. We report the results for all benchmark bandit algorithms in Table 1 (Appendix D).

### 6.3 Supplementary Experiments

In Appendix D, we conduct additional experiments to validate our policies. We focus on two areas. In Appendix D.1, we study more challenging bandit problems. In particular, we increase horizons to $n = 1\,000$ rounds, increase the number of arms to $K = 10$, vary $\mathcal{P}$, and also experiment with beta noise. We confirm all findings from Figure 2c. We also find that tuned `SoftElim` can outperform the Gittins index by a large margin, in the instances where the Gittins index needs to be approximated. In Appendix D.2, we study the robustness of tuned `SoftElim` policies. We focus on two aspects of this problem. First, we show that the quality of the policies does not deteriorate significantly as we increase the horizon $n$ and decrease the batch size $m$ in `GradBand`. Second, we study the robustness of the policies to prior misspecification.

### 6.4 RNN Policies

Our preliminary experiments showed that learning of RNN policies (Section 5.3) over long horizons ($n = 200$) is challenging if we use our variance reduction baselines (Section 6.1) alone. To mitigate this, we propose the use of *curriculum learning* [15] to further reduce variance. The key idea is to apply `GradBand` successively to problems with increasing horizons. In this experiment, we consider a simple instance of this idea with two horizons: $n' = 20$ and $n = 200$. First, we optimize the RNN policy using `GradBand` at horizon $n'$. Then we take the learned policy and use it as the initial policy for `GradBand` optimization at horizon $n$. The number of `GradBand` iterations is $L = 1\,000$. We did not make any attempt to optimize this scheme.

Results from the second optimization phase are reported in Figure 3. Figure 3a shows learning of an RNN policy for the problem in Section 6.2. The RNN policy outperforms both `TS` and the Gittins index. This does not contradict theory, as the Gittins index is not Bayes optimal in this problem. In Figure 3b, we consider a variant of this problem where arm means are drawn i.i.d. from $\text{Beta}(1, 1)$. The Gittins index is Bayes optimal in this problem, and so our learned RNN policy naturally does not outperform it. Nevertheless, it has a lower regret than `TS`.

In the final experiment, we have a $K$-armed Bayesian bandit with Bernoulli rewards. The prior $\mathcal{P}$ is over two problem instances, $\mu = (0.6, 0.9, 0.7, 0.7, \dots, 0.7)$ and $\mu = (0.2, 0.7, 0.9, 0.7, \dots, 0.7)$, which are equally likely. This problem has an interesting structure. The problem instance, and thus the optimal arm, can be easily identified by pulling arm 1. Arms 4 and beyond are *distractors*. Our RNN policies do not learn this structure; but they learn another strategy specialized to this problem. The strategy pulls only arms 2 or 3, since these are the only arms that be optimal. Thus, the RNN successfully learns to ignore the distractors. As a result, the Bayes regret of our policies (Figure 3c) does not increase with $K$. This would not happen with classic bandit algorithms.

# 7  Related Work

The regret of bandit algorithms can be reduced by tuning [58, 40, 32, 31]. None of these works used policy gradients, neural network policies, or even the sequential structure of $n$-round rewards. Duan et al. [24] optimized a similar policy to Section 5.3 using an existing optimizer. This work does not formalize the objective clearly, relates it to Bayesian bandits, or studies policies that are provably sound (Theorem 1 and Section 5.2). Silver et al. [50] applied policy gradients to a continuous bandit problem with a quadratic cost function. Since their cost is convex in arms, this exploration problem is easier than with discrete arms.

Policy gradients in RL were proposed by Williams [60], including the idea of baseline subtraction. Other early works on this topic are Sutton et al. [53] and Baxter and Bartlett [13]. Policy gradients tend to have a high variance and reducing it is an important research area [29, 46, 64, 23, 39]. Our baselines differ from those in RL, in particular because our number of states $H_t$ is not small. The baseline $b^{\mathrm{OPT}}$ uses the fact that we have a bandit problem, and thus the best arm in hindsight. Both $b^{\mathrm{OPT}}$ and $b^{\mathrm{SELF}}$ use the fact that we have access to all rewards, even of arms not pulled by policy $\pi$.

Our approach is an instance of meta-learning [55, 56], where we learn from a sample of tasks to perform well on tasks drawn from the same distribution [11, 12]. Meta-learning has been applied successfully in deep reinforcement learning (RL) [25, 26, 44]. Sequential multitask learning [17] was studied in multi-armed bandits by Azar et al. [10] and in contextual bandits by Deshmukh et al. [22]. In comparison, our setting is offline. A general template for sequential meta-learning was presented in Ortega et al. [47]. This work is conceptual and does not study policy gradients.

Maillard [41] proposed `SoftElim` with $w = 1$ and bounded the number of pulls of a suboptimal arm in Theorem 1.10. The bound has a large $O(K\Delta^{-4})$ constant, which does not seem easy to eliminate. We introduce $w$ and have a tighter analysis (Theorem 4) with a $O(1)$ constant, for $w = 8$. Also note that `SoftElim` is not very competitive with `TS` without tuning. Therefore, this approach have not received much attention in the past, and this is the first work that makes it practical. The design of `SoftElim` resembles Boltzmann exploration [52, 18] and `Exp3` (Section 5.1). The key difference is in how $S_{i,t}$ is chosen. In `Exp3` and Boltzmann exploration, $S_{i,t}$ only depends on the history of arm $i$. In `SoftElim`, $S_{i,t}$ depends on all arms and makes `SoftElim` sufficiently optimistic.

# 8  Conclusions

We take first steps towards understanding policy-gradient optimization of bandit policies. Our work addresses two main challenges of this problem. First, we derive the reward gradient of optimized policies and show how to estimate it efficiently from a sample. Second, we propose differentiable bandit policies that can outperform state-of-the-art baselines after optimization. Our experiments highlight the simplicity and generality of our approach. We also show that neural network policies can learn interesting biases.

Our goal was to clearly show the benefits of learning to explore over the state of the art. Therefore, we focus on non-contextual problems, where the optimal policy can sometimes be computed and Thompson sampling is the state of the art. Our results can be extended to structured problems. An insight that permits the generalization of `SoftElim` is that $S_{i,t}$ in (5) is a ratio of two quantities, the squared empirical suboptimality gap of arm $i$ and the variance of the mean reward estimate of arm $i$, $1/T_{i,t-1}$. Such quantities can be computed in linear bandits, for instance; and we hope to extend our work to even more general structures [34, 33, 19, 21, 57, 62]. Recently, Kveton et al. [35] and Yang and Toni [61] extended this work to contextual bandits, and Kveton et al. [35] and Min et al. [43] proposed differentiable Thompson sampling; thus resolving the open question in Section 5.

We leave open several questions of interest. First, we observe that the variance of empirical reward gradients can be high, especially in RNN policies. So any progress in variance reduction would be of a great importance. Second, except for Theorem 1, we are unaware of other bandit policy-instance pairs where the Bayes reward is concave in optimized parameters; and thus gradient ascent leads to optimal solutions. Our empirical results (Figure 1a) suggest that this phenomenon may be common. Finally, we believe that convergence guarantees for softmax exploration policies will be established in the future, based on recent advances in analyzing policy gradients in RL [2, 42].

## Broader Impact

We develop bandit-style exploration algorithms that can be easily trained on historical data. This fundamentally alters the current prevalent approach to bandit algorithm design, which is by theory. Bandit algorithms are often deployed in user-facing domains. However, we do not propose any new domain. So our impact on users should be limited to those that already exist. Beyond this, we are not aware of any societal consequences of our work, such as on welfare, fairness, or privacy.

## Acknowledgments and Disclosure of Funding

Csaba Szepesvári gratefully acknowledges funding from the Canada CIFAR AI Chairs Program, Amii, and NSERC.

There are no additional revenues related to this work.

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
