[Supplementary Material]

# A  Proof of Theorem 1

We show that the $n$-round Bayes reward of a randomized explore-then-commit policy in 2-armed Gaussian bandits is concave in the exploration horizon of the policy.

**Theorem 1.** *Consider a 2-armed Gaussian bandit where the reward of arm $i$ in round $t$ is $Y_{i,t} \sim \mathcal{N}(\mu_i, 1)$. Consider an* explore-then-commit *policy $\pi_h$ with parameter $h \in \mathcal{W} = [1, \lfloor n/2 \rfloor]$ that explores each arm $\bar{h} = \lfloor h \rfloor + Z$ times for $Z \sim \mathrm{Ber}(h - \lfloor h \rfloor)$. Then for any prior distribution $\mathcal{P}$ over arm means $\mu \in \mathbb{R}^2$, we have that $r(n; \pi_h)$ is concave in $h$.*

*Proof.* We start with the *explore-then-commit* policy [37], which is parameterized by $h \in [\lfloor n/2 \rfloor]$ and works as follows. In the first $2h$ rounds, it explores and pulls each arm $h$ times. Let $\hat{\mu}_{i,h}$ be the average reward of arm $i$ after $h$ pulls. Then, if $\hat{\mu}_{1,h} > \hat{\mu}_{2,h}$, arm 1 is pulled for the remaining $n - 2h$ rounds. Otherwise arm 2 is pulled.

Fix any problem instance $P \sim \mathcal{P}$. Without loss of generality, let arm 1 be optimal, that is $\mu_1 > \mu_2$. Let $\Delta = \mu_1 - \mu_2$. The key observation is that the expected $n$-round reward in problem instance $P$ has a closed form

$$r(n, P; \pi_h) = \mu_1 n - \Delta \left[ h + \mathbb{P}\left( \hat{\mu}_{1,h} < \hat{\mu}_{2,h} \right)(n - 2h) \right], \tag{6}$$

where

$$\mathbb{P}\left( \hat{\mu}_{1,h} < \hat{\mu}_{2,h} \right) = \mathbb{P}\left( \hat{\mu}_{1,h} - \hat{\mu}_{2,h} < 0 \right) = \mathbb{P}\left( \hat{\mu}_{1,h} - \hat{\mu}_{2,h} - \Delta < -\Delta \right)$$

$$= \Phi\left( -\Delta\sqrt{h/2} \right) = \frac{1}{\sqrt{2\pi}} \int_{x=-\infty}^{-\Delta\sqrt{h/2}} e^{-\frac{x^2}{2}} \, \mathrm{d}x \tag{7}$$

is the probability of committing to a suboptimal arm after the exploration phase. The third equality is from the fact that $\hat{\mu}_{1,h} - \hat{\mu}_{2,h} - \Delta \sim \mathcal{N}(0, 2/h)$, where $\Phi(x)$ is the cumulative distribution function of the standard normal distribution.

Our goal is to prove that $r(n, P; \pi_h)$ is concave in $h$. We rely on the following property of convex functions of a single parameter $x$. Let $f(x)$ and $g(x)$ be non-negative, decreasing, and convex in $x$. Then $f(x)g(x)$ is non-negative, decreasing, and convex in $x$. This follows from

$$(f(x)g(x))' = f'(x)g(x) + f(x)g'(x),$$
$$(f(x)g(x))'' = f''(x)g(x) + 2f'(x)g'(x) + f(x)g''(x).$$

It is easy to see that (7) is non-negative, decreasing, and convex in $h$. The same is true for $n - 2h$, under our assumption that $h \in [\lfloor n/2 \rfloor]$. As a result, $\mathbb{P}\left( \hat{\mu}_{1,h} < \hat{\mu}_{2,h} \right)(n - 2h)$ is convex in $h$, and so is $\Delta[h + \mathbb{P}\left( \hat{\mu}_{1,h} < \hat{\mu}_{2,h} \right)(n - 2h)]$. Therefore, (6) is concave in $h$. Finally, the Bayes reward is concave in $h$ because $r(n; \pi_h) = \mathbb{E}\left[ r(n, P; \pi_h) \right]$.

The last remaining issue is that parameter $h$ in the explore-then-commit policy cannot be optimized by GradBand, as it is discrete. To allow for optimization, we extend the explore-then-commit policy to continuous $h$ by randomized rounding.

The *randomized explore-then-commit* policy is parameterized by continuous $h \in [1, \lfloor n/2 \rfloor]$. The discrete $\bar{h}$ is chosen as $\bar{h} = \lfloor h \rfloor + Z$, where $Z \sim \mathrm{Ber}(h - \lfloor h \rfloor)$. Then we execute the original policy with $\bar{h}$. The key property of the randomized policy is that its $n$-round Bayes reward is a piecewise linear interpolation of that of the original policy,

$$(\lceil h \rceil - h)\, r(n; \pi_{\lfloor h \rfloor}) + (h - \lfloor h \rfloor)\, r(n; \pi_{\lceil h \rceil}).$$

By definition, the above function is continuous and concave in $h$. This concludes the proof. $\square$

# B  Gradient Proofs

**Theorem 2.** *Let $b_t : [K]^{t-1} \times [0,1]^{K \times n} \to \mathbb{R}$ be any function of previous $t-1$ pulled arms and all realized rewards, for any round $t \in [n]$. Then*

$$\nabla_w r(n; \pi_w) = \sum_{t=1}^{n} \mathbb{E}\left[ \nabla_w \log \pi_w(I_t \mid H_t) \left( \sum_{s=t}^{n} Y_{I_s,s} - b_t(I_{1:t-1}, Y) \right) \right].$$

*Proof.* The proof has two parts. First, we show that

$$\nabla_w r(n; \pi_w) = \sum_{t=1}^{n} \mathbb{E}\left[ \nabla_w \log \pi_w(I_t \mid H_t) \sum_{s=t}^{n} Y_{I_s,s} \right]. \tag{8}$$

The $n$-round Bayes reward can be expressed as $r(n; \pi_w) = \mathbb{E}\left[\mathbb{E}\left[\sum_{t=1}^{n} Y_{I_t,t} \mid Y\right]\right]$, where the outer expectation is over instances $P$ and their realized rewards $Y$, which are independent of $w$. Thus

$$\nabla_w r(n; \pi_w) = \mathbb{E}\left[ \sum_{t=1}^{n} \nabla_w \mathbb{E}\left[ Y_{I_t,t} \mid Y \right] \right].$$

In the inner expectation, only the pulled arms are random. Therefore, for any $t \in [n]$, we have

$$\mathbb{E}\left[ Y_{I_t,t} \mid Y \right] = \sum_{i_{1:t}} \mathbb{P}\left( I_{1:t} = i_{1:t} \mid Y \right) Y_{i_t,t}.$$

The key to our derivations is that the joint probability distribution over pulled arms in the first $t$ rounds, conditioned on $Y$, decomposes by the chain rule of probabilities as

$$\mathbb{P}\left( I_{1:t} = i_{1:t} \mid Y \right) = \prod_{s=1}^{t} \mathbb{P}\left( I_s = i_s \mid I_{1:s-1} = i_{1:s-1}, Y \right). \tag{9}$$

Since the policy does not act based on future rewards, we have for any $s \in [n]$ that

$$\mathbb{P}\left( I_s = i_s \mid I_{1:s-1} = i_{1:s-1}, Y \right) = \pi_w(i_s \mid i_{1:s-1}, Y_{i_1,1}, \ldots, Y_{i_{s-1},s-1}). \tag{10}$$

Finally, we use that $\nabla_w f(w) = f(w) \nabla_w \log f(w)$ holds for any non-negative differentiable $f$. This identity is known as the *score-function identity* [5] and is the basis of all policy-gradient methods. We apply it to $\mathbb{E}\left[ Y_{I_t,t} \mid Y \right]$ and obtain

$$\begin{aligned}
\nabla_w \mathbb{E}\left[ Y_{I_t,t} \mid Y \right] &= \sum_{i_{1:t}} Y_{i_t,t} \nabla_w \mathbb{P}\left( I_{1:t} = i_{1:t} \mid Y \right) \\
&= \sum_{i_{1:t}} Y_{i_t,t} \mathbb{P}\left( I_{1:t} = i_{1:t} \mid Y \right) \nabla_w \log \mathbb{P}\left( I_{1:t} = i_{1:t} \mid Y \right) \\
&= \sum_{s=1}^{t} \mathbb{E}\left[ Y_{I_t,t} \nabla_w \log \pi_w(I_s \mid H_s) \mid Y \right],
\end{aligned}$$

where the last equality follows from (9) and (10). Now we chain all equalities to obtain the *reward gradient*

$$\nabla_w r(n; \pi_w) = \sum_{t=1}^{n} \sum_{s=1}^{t} \mathbb{E}\left[ Y_{I_t,t} \nabla_w \log \pi_w(I_s \mid H_s) \right] = \sum_{t=1}^{n} \mathbb{E}\left[ \nabla_w \log \pi_w(I_t \mid H_t) \sum_{s=t}^{n} Y_{I_s,s} \right].$$

This concludes the first part of the proof.

Now we argue that $b_t$ does not change anything. Since $b_t$ depends only on $I_{1:t-1}$ and $Y$,

$$\mathbb{E}\left[ b_t(I_{1:t-1}, Y) \nabla_w \log \pi_w(I_t \mid H_t) \right] = \mathbb{E}\left[ b_t(I_{1:t-1}, Y) \mathbb{E}\left[ \nabla_w \log \pi_w(I_t \mid H_t) \mid I_{1:t-1}, Y \right] \right].$$

Now note that

$$\mathbb{E}\left[\nabla_w \log \pi_w(I_t \mid H_t) \mid I_{1:t-1}, Y\right] = \sum_{i=1}^{K} \mathbb{P}\left(I_t = i \mid I_{1:t-1}, Y\right) \nabla_w \log \pi_w(i \mid H_t)$$

$$= \sum_{i=1}^{K} \pi_w(i \mid H_t) \nabla_w \log \pi_w(i \mid H_t)$$

$$= \nabla_w \sum_{i=1}^{K} \pi_w(i \mid H_t) = 0 \, .$$

The last equality follows from $\sum_{i=1}^{K} \pi_w(i \mid H_t) = 1$, which is a constant independent of $w$. This concludes the proof. □

**Lemma 3.** *Define $\pi_{i,t}$ as in* (4)*. Let $\eta = w/K$, $V_{i,t} = \exp[wS_{i,t}/K]$, and $V_t = \sum_{j=1}^{K} V_{j,t}$. Then*

$$\nabla_w \log \pi_{i,t} = \frac{1}{\pi_{i,t}} \left[ \frac{V_{i,t}}{V_t} \left[ (1-w) \left[ \frac{S_{i,t}}{K} - \sum_{j=1}^{K} \frac{V_{j,t}}{V_t} \frac{S_{j,t}}{K} \right] - 1 \right] + \frac{1}{K} \right] \, .$$

*Proof.* First, we express the derivative of $\log \pi_{i,t}$ with respect to $w$ as

$$\nabla_w \log \pi_{i,t} = \frac{1}{\pi_{i,t}} \nabla_w \pi_{i,t} = \frac{1}{\pi_{i,t}} \left[ (1-w)\nabla_w \frac{V_{i,t}}{V_t} - \frac{V_{i,t}}{V_t} + \frac{1}{K} \right] \, .$$

Conditioned on the history, $S_{i,t}$ is a constant independent of $w$, and thus we have

$$\nabla_w \frac{V_{i,t}}{V_t} = \frac{1}{V_t} \nabla_w V_{i,t} + V_{i,t} \nabla_w \frac{1}{V_t} = \frac{V_{i,t}S_{i,t}}{V_tK} - \frac{V_{i,t}}{V_t^2} \sum_{j=1}^{K} V_{j,t} \frac{S_{j,t}}{K}$$

$$= \frac{V_{i,t}}{V_t} \left[ \frac{S_{i,t}}{K} - \sum_{j=1}^{K} \frac{V_{j,t}}{V_t} \frac{S_{j,t}}{K} \right] \, .$$

This concludes the proof. □

# C  Analysis of `SoftElim`

We informally justify `SoftElim` in Appendix C.1. Then we bound its regret in Appendix C.2.

## C.1  Informal Analysis

We start with an informal argument in a 2-armed bandit. Let arm 1 be optimal, that is $\mu_1 > \mu_2$. Let $\Delta = \mu_1 - \mu_2$ be the gap. Fix any round $t$ by which arm 2 has been pulled "often", so that we get $T_{2,t-1} = \Omega(\Delta^{-2} \log n)$ and $\hat{\mu}_{2,t-1} \leq \mu_2 + \Delta/3$ with a high probability. Let

$$\hat{\mu}_{\max,t} = \max\{\hat{\mu}_{1,t}, \hat{\mu}_{2,t}\} \ .$$

Now consider two cases. First, when $\hat{\mu}_{\max,t-1} = \hat{\mu}_{1,t-1}$, arm 1 is pulled with probability of at least 0.5, by the definition of $\pi_{1,t}$. Second, when $\hat{\mu}_{\max,t-1} = \hat{\mu}_{2,t-1}$, we have

$$\pi_{1,t} = \exp[-2(\hat{\mu}_{2,t-1} - \hat{\mu}_{1,t-1})^2 T_{1,t-1}]\pi_{2,t} \geq \exp[-2(\mu_1 - \hat{\mu}_{1,t-1})^2 T_{1,t-1}]\pi_{2,t} \ ,$$

where the last inequality follows from $\hat{\mu}_{1,t-1} \leq \hat{\mu}_{2,t-1} \leq \mu_2 + \Delta/3 \leq \mu_1$ and holds with a high probability. This means that arm 1 is pulled "sufficiently often" relative to arm 2, proportionally to the deviation of $\hat{\mu}_{1,t-1}$ from $\mu_1$. Therefore, `SoftElim` eventually enters a regime where arm 1 has been pulled "often", so that $T_{1,t-1} = \Omega(\Delta^{-2} \log n)$ and $\hat{\mu}_{1,t-1} \geq \mu_1 - \Delta/3$ holds with a high probability. Then both $S_{1,t} = 0$ and $S_{2,t} = \Omega(\log n)$ hold with a high probability, and arm 2 is unlikely to be pulled.

## C.2  Regret Bound

We bound the $n$-round regret of `SoftElim` below.

**Theorem 4.** *Let $P$ be any $K$-armed bandit problem where arm 1 is optimal, that is $\mu_1 > \max_{i>1} \mu_i$. Let $\Delta_i = \mu_1 - \mu_i$ and $w = \sqrt{8}$. Then $R(n, P; \pi_w) \leq \sum_{i=2}^{K}(2e + 1)\left(16\Delta_i^{-1} \log n + \Delta_i\right) + 5\Delta_i$.*

*Proof.* Each arm is initially pulled once. Therefore,

$$R(n, P; \pi_w) = \sum_{i=2}^{K} \Delta_i \left( \sum_{t=K+1}^{n} \mathbb{P}\left(I_t = i\right) + 1 \right) \ .$$

Now we decompose the probability of pulling each arm $i$ as

$$\sum_{t=K+1}^{n} \mathbb{P}\left(I_t = i\right) = \sum_{t=K+1}^{n} \mathbb{P}\left(I_t = i, T_{i,t-1} \leq m\right) +$$
$$\sum_{t=K+1}^{n} \mathbb{P}\left(I_t = i, T_{i,t-1} > m, T_{1,t-1} \leq m\right) +$$
$$\sum_{t=K+1}^{n} \mathbb{P}\left(I_t = i, T_{i,t-1} > m, T_{1,t-1} > m\right) \ ,$$

where $m$ is chosen later. In the rest of the proof, we bound each above term separately. To simplify notation, use $\gamma = 1/w^2$ in instead of $w^2$.

## C.3  Upper Bound on Term 1

Fix suboptimal arm $i$. Since $T_{i,t} = T_{i,t-1} + 1$ on event $I_t = i$ and arm $i$ is initially pulled once, we have

$$\sum_{t=K+1}^{n} \mathbb{P}\left(I_t = i, T_{i,t-1} \leq m\right) \leq m - 1 \ . \tag{11}$$

## C.4 Upper Bound on Term 3

Fix suboptimal arm $i$ and round $t$. Let

$$E_{1,t} = \left\{ \hat{\mu}_{1,t-1} > \mu_1 - \frac{\Delta_i}{4} \right\}, \quad E_{i,t} = \left\{ \hat{\mu}_{i,t-1} < \mu_i + \frac{\Delta_i}{4} \right\},$$

be the events that empirical means of arms 1 and $i$, respectively, are "close" to their means. Then

$$\begin{aligned}
\mathbb{P}\left(I_t = i, T_{i,t-1} > m, T_{1,t-1} > m\right) \\
\leq \mathbb{P}\left(I_t = i, T_{i,t-1} > m, E_{1,t}\right) + \mathbb{P}\left(\bar{E}_{1,t}, T_{1,t-1} > m\right) \\
\leq \mathbb{P}\left(I_t = i, T_{i,t-1} > m, E_{1,t}, E_{i,t}\right) + \mathbb{P}\left(\bar{E}_{1,t}, T_{1,t-1} > m\right) + \mathbb{P}\left(\bar{E}_{i,t}, T_{i,t-1} > m\right).
\end{aligned}$$

Let $m = \lceil 16\Delta_i^{-2} \log n \rceil$. By the union bound and Hoeffding's inequality, we get

$$\mathbb{P}\left(\bar{E}_{1,t}, T_{1,t-1} > m\right) \leq \sum_{s=m+1}^{n} \mathbb{P}\left(\mu_1 - \hat{\mu}_{1,t-1} \geq \frac{\Delta_i}{4}, T_{1,t-1} = s\right) < n \exp\left[-2\frac{\Delta_i^2}{16}m\right] = n^{-1},$$

$$\mathbb{P}\left(\bar{E}_{i,t}, T_{i,t-1} > m\right) \leq \sum_{s=m+1}^{n} \mathbb{P}\left(\hat{\mu}_{i,t-1} - \mu_i \geq \frac{\Delta_i}{4}, T_{i,t-1} = s\right) < n \exp\left[-2\frac{\Delta_i^2}{16}m\right] = n^{-1}.$$

It follows that

$$\mathbb{P}\left(I_t = i, T_{i,t-1} > m, T_{1,t-1} > m\right) \leq \mathbb{P}\left(I_t = i, T_{i,t-1} > m, E_{1,t}, E_{i,t}\right) + 2n^{-1}.$$

Now note that $\hat{\mu}_{1,t-1} - \hat{\mu}_{i,t-1} \geq \Delta_i/2$ on events $E_{1,t}$ and $E_{i,t}$. Let

$$\hat{\mu}_{\max,t-1} = \max_{i \in [K]} \hat{\mu}_{i,t-1} \tag{12}$$

be the highest empirical mean in round $t$. Since $\hat{\mu}_{\max,t-1} \geq \hat{\mu}_{1,t-1}$, we have $\hat{\mu}_{\max,t-1} - \hat{\mu}_{i,t-1} \geq \Delta_i/2$. Therefore, on event $T_{i,t-1} > m$, we get

$$p_{i,t} \leq \exp[-2\gamma(\hat{\mu}_{\max,t-1} - \hat{\mu}_{i,t-1})^2 T_{i,t-1}] \leq \exp\left[-2\gamma\frac{\Delta_i^2}{4}m\right] \leq n^{-8\gamma}. \tag{13}$$

Finally, we chain all inequalities over all rounds and get that term 3 is bounded as

$$\sum_{t=K+1}^{n} \mathbb{P}\left(I_t = i, T_{i,t-1} > m, T_{1,t-1} > m\right) \leq n^{1-8\gamma} + 2. \tag{14}$$

## C.5 Upper Bound on Term 2

Fix suboptimal arm $i$ and round $t$. First, we apply Hoeffding's inequality to arm $i$, as in Appendix C.4, and get

$$\begin{aligned}
\mathbb{P}\left(I_t = i, T_{i,t-1} > m, T_{1,t-1} \leq m\right) &\leq \mathbb{P}\left(I_t = i, T_{i,t-1} > m, T_{1,t-1} \leq m, E_{i,t}\right) + n^{-1} \\
&= \mathbb{E}\left[p_{i,t}\mathbb{1}\{T_{i,t-1} > m, T_{1,t-1} \leq m, E_{i,t}\}\right] + n^{-1}.
\end{aligned}$$

Let $\hat{\mu}_{\max,t-1}$ be defined as in (12). Now we bound $p_{i,t}$ from above using $p_{1,t}$. We consider two cases. First, suppose that $\hat{\mu}_{\max,t-1} > \mu_1 - \Delta_i/4$. Then we have (13). On the other hand, when $\hat{\mu}_{\max,t-1} \leq \mu_1 - \Delta_i/4$, we have

$$p_{i,t} = \frac{\exp[-2\gamma(\hat{\mu}_{\max,t-1} - \hat{\mu}_{i,t-1})^2 T_{i,t-1}]}{\exp[-2\gamma(\hat{\mu}_{\max,t-1} - \hat{\mu}_{1,t-1})^2 T_{1,t-1}]}p_{1,t} \leq \exp[2\gamma(\mu_1 - \hat{\mu}_{1,t-1})^2 T_{1,t-1}]p_{1,t}. \tag{15}$$

It follows that

$$p_{i,t} \leq \exp[2\gamma(\mu_1 - \hat{\mu}_{1,t-1})^2 T_{1,t-1}]p_{1,t} + n^{-8\gamma},$$

and we further get that

$$\begin{aligned}
\mathbb{E}\left[p_{i,t}\mathbb{1}\{T_{i,t-1} > m, T_{1,t-1} \leq m, E_{i,t}\}\right] \\
\leq \mathbb{E}\left[\exp[2\gamma(\mu_1 - \hat{\mu}_{1,t-1})^2 T_{1,t-1}]p_{1,t}\mathbb{1}\{T_{1,t-1} \leq m\}\right] + n^{-8\gamma} \\
= \mathbb{E}\left[\exp[2\gamma(\mu_1 - \hat{\mu}_{1,t-1})^2 T_{1,t-1}]\mathbb{1}\{I_t = 1, T_{1,t-1} \leq m\}\right] + n^{-8\gamma}.
\end{aligned}$$

With a slight abuse of notation, let $\hat{\mu}_{1,s}$ denote the average reward of arm $1$ after $s$ pulls. Then, since $T_{1,t} = T_{1,t-1} + 1$ on event $I_t = 1$, we have

$$\sum_{t=K+1}^{n} \mathbb{E}\left[\exp[2\gamma(\mu_1 - \hat{\mu}_{1,t-1})^2 T_{1,t-1}]\mathbb{1}\{I_t = 1, T_{1,t-1} \leq m\}\right] \leq \sum_{s=1}^{m} \mathbb{E}\left[\exp[2\gamma(\mu_1 - \hat{\mu}_{1,s})^2 s]\right].$$

Now fix the number of pulls $s$ and note that

$$\mathbb{E}\left[\exp[2\gamma(\mu_1 - \hat{\mu}_{1,s})^2 s]\right] \leq \sum_{\ell=0}^{\infty} \mathbb{P}\left(\frac{\ell+1}{\sqrt{s}} > |\mu_1 - \hat{\mu}_{1,s}| \geq \frac{\ell}{\sqrt{s}}\right) \exp[2\gamma(\ell+1)^2]$$

$$\leq \sum_{\ell=0}^{\infty} \mathbb{P}\left(|\mu_1 - \hat{\mu}_{1,s}| \geq \frac{\ell}{\sqrt{s}}\right) \exp[2\gamma(\ell+1)^2]$$

$$\leq 2 \sum_{\ell=0}^{\infty} \exp[2\gamma(\ell+1)^2 - 2\ell^2],$$

where the last step is by Hoeffding's inequality. The above sum can be easily bounded for any $\gamma < 1$. In particular, for $\gamma = 1/8$, the bound is

$$\sum_{\ell=0}^{\infty} \exp\left[\frac{(\ell+1)^2}{4} - 2\ell^2\right] \leq e^{\frac{1}{4}} + \sum_{\ell=1}^{\infty} 2^{-\ell} \leq e.$$

Now we combine all above inequalities and get that term 2 is bounded as

$$\sum_{t=K+1}^{n} \mathbb{P}\left(I_t = i, T_{i,t-1} > m, T_{1,t-1} \leq m\right) \leq 2em + n^{1-8\gamma} + 1. \tag{16}$$

Finally, we chain (11), (14), and (16); and use that $m \leq 16\Delta_i^{-2}\log n + 1$. $\qquad\square$

Figure 4: The Bayes regret of `Exp3` and `SoftElim` policies, as a function of `GradBand` iterations. We report the average over 20 runs.

Figure 5: Robustness to batch size $m$, horizon $n$, and prior misspecification.

## D  Supplementary Experiments

We conduct additional experiments in this section.

### D.1  More Complex Problems

Now we apply `GradBand` to three additional problems. The first problem is a beta bandit, where the rewards of arm $i$ are drawn from $\text{Beta}(v\mu_i, v(1 - \mu_i))$ and $v = 4$ controls the variance of rewards. The remaining parameters of the problem are set as in the Bernoulli bandit in Section 6.2. The remaining two problems are variants of Bernoulli and beta bandits, where the number of arms is $K = 10$, their mean rewards are drawn as $\mu_i \sim \text{Beta}(1, 1)$, and the horizon is $n = 1\,000$. These problems are more challenging variants of our earlier problems, with 2 arms and a fixed gap.

The regret of our policies is reported in Figure 4. In all problems, the regret of `SoftElim` is lower than that of `TS`, which is a highly competitive baseline. The most significant improvements are in beta bandits, where `SoftElim` adapts to the lower variance of rewards. The poor performance of `TS` is due to the Bernoulli rounding in the algorithm (Section 6.2), which replaces low-variance beta rewards with high-variance Bernoulli rewards. As observed earlier, tuned `Exp3` is not competitive.

### D.2  Robustness to Model and Algorithm Parameters

This section presents three experiments, which show the robustness of `SoftElim` to the setting of its parameters and model misspecification. These experiments are conducted on the larger Bernoulli and beta problems in Appendix D.1.

In Figure 5a, we report the $n$-round regret of tuned `SoftElim` as a function of batch size $m$ in `GradBand`. We observe that the mean regret, across all runs, is relatively stable as we decrease the batch size from 1000 to 100. The variance increases though. Setting $m = 100$ reduces the run time of `GradBand` ten fold, when compared to $m = 1\,000$ used in our earlier experiments.

In Figure 5b, we report the $n$-round regret of tuned `SoftElim` as we vary the horizon $n$, from 200 to 2 000 rounds. We observe that the regret is roughly linear in $\log n$. This scaling is theoretically optimal. We expect it when the variance of gradients does not dominate `GradBand`, and thus `GradBand` can equally well optimize policies at shorter and longer horizons.

| Figure | 2c | 4a | 4b | 4c | 3a | 3b |
|---|---|---|---|---|---|---|
| Gittins index | $3.89 \pm 0.07$ | $3.89 \pm 0.07$ | x | x | $3.89 \pm 0.07$ | $2.26 \pm 0.04$ |
| TS | $5.47 \pm 0.05$ | $5.47 \pm 0.05$ | $28.06 \pm 0.45$ | $28.06 \pm 0.45$ | $5.47 \pm 0.05$ | $3.50 \pm 0.03$ |
| UCB1 | $9.95 \pm 0.03$ | $9.95 \pm 0.03$ | $129.09 \pm 0.60$ | $129.09 \pm 0.60$ | $9.95 \pm 0.03$ | $8.52 \pm 0.03$ |
| UCB-V | $15.79 \pm 0.03$ | | $289.82 \pm 1.90$ | $276.07 \pm 1.65$ | $15.79 \pm 0.03$ | $19.03 \pm 0.10$ |

Table 1: The Bayes regret of baseline bandit algorithms in Figures 2, 3 and 4. The crosses mark computationally-prohibitive experiments.

In Figure 5c, we investigate the robustness of tuned `SoftElim` to prior misspecification. In particular, we tune `SoftElim` on a Bernoulli bandit where $\mu_i \sim \text{Beta}(\alpha, 11 - \alpha)$ for $\alpha \in [10]$ and measure its $n$-round regret on a Bernoulli bandit with another $\alpha$. We observe that the regret increases when we train and test on different problems. For instance, when we train and test on $\mu_i \sim \text{Beta}(1, 9)$, the regret is about 30. However, when we train on $\mu_i \sim \text{Beta}(9, 1)$, the regret increases to about 50. Nevertheless, we do not observe catastrophic failures, which would happen if the regret increased by an order of magnitude. We conclude that `GradBand` is relatively robust to prior misspecification.

# E  RNN Implementation

We carry out the RNN experiments using PyTorch framework. In this paper, we restrict ourselves to binary 0/1 rewards. For all experiments, our policy network is a single layer LSTM followed by LeakyRELU non-linearity and a fully connected layer. We use the fixed LSTM latent state dimension of 50, irrespective of numbers of arms. The implementation of the policy network is provided in the code snippet below:

```python
class RecurrentPolicyNet(nn.Module):
  def __init__(self, K=2, d=50):
    super(RecurrentPolicyNet, self).__init__()
    self.action_size = K  # Number of arms
    self.hidden_size = d
    self.input_size = 2*d

    self.arm_emb = nn.Embedding(K, self.hidden_size)      # Number of arms
    self.reward_emb = nn.Embedding(2, self.hidden_size)  # For 0 reward
    or 1 reward
    self.rnn = nn.LSTMCell(input_size=self.input_size,
                           hidden_size=self.hidden_size)
    self.relu = nn.LeakyReLU()
    self.linear = nn.Linear(self.hidden_size, self.action_size)

    self.hprev = None

  def reset(self):
    self.hprev = None

  def forward(self, action, reward):
    arm = self.arm_emb(action)
    rew = self.rew_emb(reward)

    inp = torch.cat((arm, rew), 1)
    h = self.rnn(inp, self.hprev)
    self.hprev = h

    h = self.relu(h[0])
    y = self.linear(h)

    return y
```

Listing 1: Policy Network

To train the policy we use the proposed `GradBand` algorithm as presented in Alg. 1. We used a batch-size $m = 500$ for all experiments. Along with theoretically motivated steps, we had to apply a few practical tricks:

- Instead of SGD, we used adaptive optimizers like Adam or Yogi [63].
- We used an exponential decaying learning rate schedule. We start with a learning rate of 0.001 and decay every step by a factor of 0.999.
- We used annealing over the probability to play an arm. This encourages exploration in early phase of training. In particular we used temperature $= 1/(1 - \exp(-5i/L))$, where $i$ is current training iteration and $L$ is the total number of training iterations.
- We applied curriculum learning as described in Section 6.4.

Our training procedure is highlighted in the code snippet below.

```python
optimizer = torch.optim.Adam(policy.parameters(), lr=0.001)
scheduler = torch.optim.lr_scheduler.ExponentialLR(optimizer, 0.999)

...

probs = rnn_policy_network(previous_action, previous_reward)
```

```
 7 m = Categorical(probs/temperature)      # probability over K arms with
       temperature
 8 action = m.sample()                     # select one arm
 9 reward = bandit.play(action)            # receive reward
10
11 ...
12
13 loss = -m.log_prob(action) * (cummulative_reward - baseline)   # Eq (3)
14 loss.backward()     # Eq (9)
15 optimizer.step()
16 scheduler.step()
17
18 ...
```

Listing 2: Training overview