[Reviews · NeurIPS 2020]

Review 1

Summary and Contributions: The authors consider meta-learning in Bayesian bandits, where the learner observes bandit problems drawn from some unknown distribution. The focus is on differentiable policies, whose parameters are meta-learned by gradient methods on these sampled instances to minimize the expected Bayesian regret. A novel differentiable bandit strategy (SoftElim) is proposed which enjoys sub-linear O(\log(n)/\Delta) regret. Experiments are conducted on simple problems and show the effectiveness of optimizing differentiable strategies compared to relevant baselines.

Strengths: The problem of meta-learning/multi-task learning/transfer learning in bandits is relevant and still it hasn't received much attention as compared to the RL literature. Towards this direction, the ideas presented here are sound and simple and could be of interest for people working in this field. The SoftElim algorithm has its own merits and could be of interest even outside the specific setting considered here.

Weaknesses: - Novelty: this kind of parametrized policies, the derivation of their gradients, and the usage of baselines for variance reduction are well established concepts in the RL/policy-search literature (and others) as acknowledged in the paper, so I did not find this part too novel. On the other hand, the specific baseline adopted here (b^{self}) seems interesting and simple. - Experiments: the experiments are conducted on very simple bandit problems. Since, from my understanding, this paper makes the attempt to design algorithms that are simple, scalable, and widely applicable (e.g., as opposed to strategies with very strong theoretical guarantees but little scalability), I was expecting experiments on more complex problems. Also, the horizon used in the experiments seems relatively small (at least compared to those typically adopted in other bandit papers) and it is not clear how the proposed method scales as a function of n. - Applicability: related to the previous point, it seems that the algorithm requires large batch-sizes to stabilize learning. Anyway, I believe that in real applications we might also be concerned with the performance during training/optimization time rather than only the asymptotics. Though converging to very good performance, the approach seems to require solving quite a lot of bandit problems before getting there, in which the regret might be much higher than that of simple bandit strategies like TS (e.g., in Fig. 2a GradBand converges in about 10 iterations, but that is equivalent to about 10k bandit problems using a batch size of m=1000). In what kind of applications do you see this kind of approach mostly applicable? (see also detailed comments/questions below)

Correctness: All methods and claims seem correct (I quickly assessed the validity of the proofs). I have only one doubt regarding Lemma 3 (see detailed comments/questions below). The empirical methodology also seems correct and I believe sufficient details have been provided to allow reproducibility.

Clarity: The paper is very well-written and easy to follow. All concepts are introduced with sufficient detail and intuition.

Relation to Prior Work: Prior works are throughly discusses, especially those on meta-learning/multi-task learning for bandits and those on policy gradients. I would also reserve some discussion for works on structured bandits (e.g, [1,2,3,4]) which also concern bandit strategies that exploit prior knowledge in general forms. In those works the prior/structure is provided to the learner, while the methods presented here learn it from samples and so might be combined with the structure-adaptive ones. [1] Lattimore, Tor, and Rémi Munos. "Bounded regret for finite-armed structured bandits." Advances in Neural Information Processing Systems. 2014. [2] Combes, Richard, Stefan Magureanu, and Alexandre Proutiere. "Minimal exploration in structured stochastic bandits." Advances in Neural Information Processing Systems. 2017. [3] Tirinzoni, Andrea, Alessandro Lazaric, and Marcello Restelli. "A Novel Confidence-Based Algorithm for Structured Bandits". Proceedings of the Twenty Third International Conference on Artificial Intelligence and Statistics. 2020. [4] Degenne, Rémy, Han Shao, and Wouter M. Koolen. "Structure Adaptive Algorithms for Stochastic Bandits.". ICML. 2020.

Reproducibility: Yes

Additional Feedback: 1. As in standard policy-gradient methods, it seems that two key parameters are the batch-size m and the horizon n. It would be good to provide some sensitivity analysis on these parameters to better assess how the approach scales to complex problems. In particular, what is the effect of the horizon on the gradient estimation? Does the variance blow up or is the baseline sufficient to keep it under control? 2. Regarding the need for large batch-sizes, I believe that in practice the performance of the algorithm during training also matters, especially if this training is performed on a large number of bandit instances. In this sense, it might be good to have differentiable strategies that are provably efficient (e.g., with sub-linear regret) for a range of parameter values, so that whather value of \theta we encounter during its optimization will not performed poorly. The SoftElim algorithm could be a good idea in this direction since it seems to enjoy sub-linear regret for at least two values of \theta. Do you think it would be possible to show that there exists an interval of values of \theta in which SoftElim enjoys sub-linear (e.g., worst-case) regret? 3. Regarding the baselines, b^{self} is simple and intuitive. I was wondering: do you have any theoretical insight on its variance reduction compared to using another baseline or no baseline at all? Also, since the gradient estimator is basically G(PO)MDP, why not computing the "optimal" baseline by directly minimizing the variance of the estimator (e.g., [1])? Is there any complication/limitation that is specific of this setting? 4. Regarding Lemma 3, S_{i,t} depends on \theta through 1/p_{i,l}, right? Was that term differentiated? I could not see it in the proof. Some minor comments: - In Sec. 2 it is mentioned that the agent knows the distribution P over bandit problems. Later it is mentioned that the assumption is not really required but it is only for simplifying exposition. I did not see where it served to simplify exposition. - Sec. 6.2 mentions Tensorflow for the implementation of GradBand, while the appendix reports Pytorch code for the RNN. Is that a mistake or were the two parts implemented with different frameworks? [1] Peters, Jan, and Stefan Schaal. "Reinforcement learning of motor skills with policy gradients." Neural networks 21.4 (2008): 682-697. UPDATE: The authors addressed most of my comments and provided further experiments to assess the algorithm's sensitivity to n/m. I have increased my score accordingly.


Review 2

Summary and Contributions: This paper proposes a form of meta-learning for learning bandit algorithms in the context of Bayesian bandits where the learning agent has access to bandit instances sampled from an unknown distribution $\cal P$. The problem is formulated as a policy-gradient optimization of the Bayes expected reward of bandit policies over $n$-round, equivalent to minimizing their $n$-round Bayes expected regret. After giving a generic iterative gradient-based meta-algorithm (GradBand), the paper derives gradient estimators and proposes variance reduction techniques. The rest of the paper illustrates the overall approach for some bandit algorithms for which policies are differentiable, including the well-known Exp3, and SoftElim, a novel soft elimination algorithm with sublinear regret, and also on a LSTM implementation of a RNN. Experiments are provided to show the flexibility of the approach

Strengths: The main strength of the paper is in providing a solid framework for understanding (and possibly using in practice) policy-gradient optimization of bandit policies.

Weaknesses: As recognized by the authors, the proposed objective is not the standard one used in bandits (it doesn't optimize reward per problem instance, but rather the average reward over all instances) and as such this may limit the use of these techniques, as average reward may not be appropriate to guard against worst-case failures.

Correctness: Didn't see any issues, although I have not looked at all the proofs in the supplementary material

Clarity: Yes

Relation to Prior Work: Yes

Reproducibility: Yes

Additional Feedback: I read the rebuttal and other reviews, and I keep my current score


Review 3

Summary and Contributions: The paper studies a Bayesian bandits problem, where the agent is assumed to have access to bandit instances (a probability distribution P), and each round in addition to the reward of the pulled arm, the agent generates a vector of rewards Y sampled from P sampled from an unknown distribution \mathcal{P} (distribution of distributions). The goal is to learn a bandit algorithm that achieves high reward (expected cumulative rewards where the expectation is taken over not only the pulled arm, but also the distribution P and rewards realization Y). An iterative gradient-based algorithm is proposed to find the optimal parameter of the policy.

Strengths: The paper gives a useful way of tuning a bandit algorithm indexed by a parameter. Closed forms of reward gradients of the proposed gradient-based algorithm are computed and can be used directly. A few types of policies that are differentiable are proposed with policy gradients calculated.

Weaknesses: 1. The overall claim (at least the first half of the paper) of the study being one of learning a bandit algorithm (as opposed to using a given bandit algorithm to learn the right actions in a given problem) is a bit misleading and somewhat overstated. In the end the study is about parameter-tuning a given class of algorithms (or a single algorithm with a tunable parameter). And if my reading of the experiments is correct (see more on this below), then they are all for the tuning of a single algorithm for a single instance of bandit. 2. The point of generalizing from $P$ to $\mathcal{P}$ is well taken, but while $P$ appears to be a joint distribution across all $K$ arms, it does not appear to be across time t. Is the assumption that rewards are independent across time? This would rule out Markov chains and other time-dependent processes as arms. 3. It is assumed that at each step the algorithm can sample m problem instances and run a policy on each in order to estimate the policy gradients. Is not clear to me what this assumption means in practice: a few application examples here would be very helpful. 4. Furthermore, it is not clear to me whether the regret from all these sampled instances are counted in the analysis. If not, then the regret analysis would be cheating, and if yes, then what really is the fundamental difference between this meta-learning and treating the overall distribution $\mathcal{P}$ as a single/ensemble bandit? 5. I fail to see the point of the experiments (see more comments below): in all three experiments, the problem seems to consist of a single bandit. The first two were said to include two bandit instances but they look identical; the third one has only one instance. ======= post rebuttal ============= The rebuttal addresses my points 2,3,4. On points 1&5, I may be misunderstanding the problem setting (not entirely I hope), but the rebuttal is not helping if that's the case. In particular, my point 5 on the experiment in 6.1 and 6.2, the rebuttal does not help at all: there are two bandit instances that are mirror images of each other, so why are they not considered the same instance -- surely we can just reindex/flip the arms, no? And in any case these arms are not pre-indexed and we just observe rewards, so for all intent and purposes these two instances would look identical to a player. I agree a single instance would not be very interesting, and that's exactly the point behind my question: why choose such an experiment?

Correctness: Appears to be.

Clarity: Could be better organized and be more consistent in the overall claim (see comments above).

Relation to Prior Work: Yes.

Reproducibility: Yes

Additional Feedback: Eqn (2): why is the best arm i_* given by expectation over Y_{i,1}, the reward in round 1? Why is this optimization not history-dependent? First equation in Section 4: it would be very helpful to elaborate on why the RHS is written this way, i.e., with the last term b_t() which is said not to affect its value or the equality, which should also mean that it could have been left out of the equation? Section 6.1, the first experiment with two bandit instances: I'm missing something here, for these two bandit problems look identical to me. They each consists of two arm, with respectively the same mean combination. Surely the ordering of the arms is inconsequential. Why are these considered "two" instances?

[Author Response · NeurIPS 2020]

Thank you for the reviews of our paper. We appreciate that you like the simplicity of our approach and see its potential impact on the bandit community. We will revise the paper accordingly. Our rebuttal is below.

**Reviewer 1**

The goal of our work was easy reproducibility and clearly showing the benefits of learning to explore over the state of the art. Therefore, we focus on non-contextual bandits, where the optimal policy (Gittins index) can be sometimes computed and Thompson sampling (TS) is the state of the art. We discuss a contextual extension in Section 8.

Your main concern seems to be how the performance of `GradBand` depends on horizons $n$ and batch sizes $m$. We observe empirically that doubling of $n$ requires doubling of $m$, to get policies of a similar quality. The run time of `GradBand` is linear in $n$ and $m$, and this currently limits what we can do. To show the robustness of our reported results, we decrease batch sizes up to $m = 100$ and increase horizons up to 5 fold.

| New horizon | $n$ | $n$ | $n$ | $n$ | $2n$ | $5n$ |
|---|---|---|---|---|---|---|
| Batch size $m$ | 100 | 200 | 500 | 1000 | 1000 | 1000 |
| 2 Bernoulli arms, $n = 200$ (Figure 2b) | $4.85 \pm 0.23$ | $4.86 \pm 0.23$ | $4.75 \pm 0.08$ | $4.75 \pm 0.05$ | $5.93 \pm 0.14$ | $6.68 \pm 0.18$ |
| 10 Bernoulli arms, $n = 1000$ (Figure 2c) | $27.36 \pm 1.35$ | $23.65 \pm 0.96$ | $23.29 \pm 0.61$ | $24.88 \pm 0.76$ | $30.97 \pm 1.66$ | $39.22 \pm 2.75$ |
| 10 beta arms, $n = 1000$ (Figure 2c) | $15.75 \pm 2.86$ | $14.38 \pm 1.99$ | $10.68 \pm 0.35$ | $10.64 \pm 0.25$ | $14.05 \pm 0.59$ | $18.36 \pm 1.02$ |

The above results are for `SoftElim` and all problems in Figure 2. We observe that the regret increases as $m$ decreases, since the gradients are more noisy. But even at $m = 100$, our policies outperform TS (Figure 2) and are computed 10 times faster than in the paper. The policies for longer horizons also perform well and outperform TS.

Feedback 1: See above.

Feedback 2: Theorem 4 is an instance-dependent upper bound on the $n$-round regret of `SoftElim`. It is proved for $\theta = 8$, which was obtained by tuning constants. An analogous bound, with worse constants, holds for any $\theta \in (1, 8]$. This can be seen in the proof in Appendix C, which only requires that $\gamma = 1/\theta \in [1/8, 1)$.

Feedback 3: Existing variance minimizing techniques are hard to apply to our problem because 1) our state space, the space of all histories, is at least exponential in $n$; and 2) the shape of the value function, the future regret as a function of history, is unknown and likely hard to approximate. The baseline $b^{\text{SELF}}$ is an independent run of bandit policy $\theta$ on the same rewards. When the policy is conservative and over-explores, two of its independent runs are likely to have similar cumulative rewards, and thus $b^{\text{SELF}}$ is a good baseline. This is how we choose the initial $\theta$ in `GradBand`.

Feedback 4: Conditioned on history $H_{t-1}$, $S_{i,t}$ is a constant independent of $\theta$. Thus the proof is correct.

**Reviewer 2**

The average case is not always limiting. For instance, a standard objective in recommender systems is to personalize well on average over users. When each user is viewed as a bandit and $\mathcal{P}$ is a distribution over them, we get our setting.

**Reviewer 3**

We believe that the reviewer misunderstood our approach. We have two learning algorithms: the bandit policy (agent) in (1), which adapts to an unknown problem instance $P \sim \mathcal{P}$ over $n$ rounds; and a meta-learner `GradBand`, which optimizes the agent by gradient ascent in $L$ iterations. The agent in (1) is a standard bandit policy, which a function of its history $H_{t-1}$ and parameters $\theta$, and does not use rewards of non-pulled arms. In each iteration, `GradBand` runs the agent $m$ times. In each run $j \in [m]$, the agent is executed on rewards $Y^j \in [0, 1]^{K \times n}$ sampled by `GradBand`, for all $K$ arms in $n$ rounds in bandit instance $P^j \sim \mathcal{P}$. The ability to sample $Y^j$ is a weaker assumption than knowing the prior $\mathcal{P}$, as in Thompson sampling. In that case, the meta-learner could sample bandit instance $P^j \sim \mathcal{P}$ and then generate all its rewards over $n$ rounds. The priors are common in practice and can be learned from historic data.

Weaknesses 1 and 5: We optimize $\theta$ in a class of bandit policies parameterized by $\theta$. In Sections 6.1 and 6.2, $\mathcal{P}$ is a distribution over two symmetric bandit instances. A single instance would be trivial, since then the optimal solution would be pulling a single arm, irrespective of the history. In Section 6.3, $\mathcal{P}$ is a distribution over bandit instances whose means are drawn independently from a beta prior. That is, there are uncountably many instances.

Weakness 2: We assume independence of rewards over round $t \in [n]$, as in stochastic bandits.

Weakness 3: See the first paragraph.

Weakness 4: `GradBand` is an offline algorithm that optimizes the Bayes reward, which a function of $\theta$. It does not have regret. Does it have any guarantee on optimizing $\theta$? In simple policy classes (Theorem 1), where the Bayes reward is concave in $\theta$, `GradBand` has the same guarantees as gradient ascent and converges to $\theta_*$. In general, the Bayes reward is non-concave in $\theta$ and only good empirical performance can be established. The regret in experiments is measured on $m$ sampled bandit instances that are independent of those used in optimization by `GradBand`. So no cheating.

[Meta-Review · NeurIPS 2020]

The rebuttal helped clarify the questions raised in the review. The consensus reached in the discussion is that this is a borderline-plus paper. The reviewers appreciate the contribution's practicality, relevance and usefulness, and at the same time they do remain concerned about the narrow scope, and would rather have seen the policy-gradient method applied to parameterized policies for more complex learning problems. On the whole, this is a worthwhile addition to the program. The rebuttal did not answer one question successfully, namely regarding the setup in the experiments section, where the learning process operating at two-levels remained confusing. Yet in my opinion the experimental setup approach is sound, and I was not confused by its description in the paper or by the rebuttal. I encourage the authors to further clarify and separate the two levels to address this in the camera ready version.